# Association between non-HDLC and 1-year prognosis in patients with spontaneous intracerebral haemorrhage: a prospective cohort study from 13 hospitals in Beijing

Yu Wang [1,2,3] Jianwei Wu,[1,2] Anxin Wang,[1,2] Ruixuan Jiang,[1,2] Xingquan Zhao [1,2,4] Wenjuan Wang[1,2]

YW and JW contributed equally.

YW and JW are joint first authors.

For numbered affiliations see end of article.

**Correspondence to**
Dr Wenjuan Wang;
tong_ttyy@163.com and
Dr Xingquan Zhao;
zxq@vip.163.com

## ABSTRACT

**Objectives** Previous studies suggested an inverse association between lipoprotein cholesterols and bleeding risk, while limited data were available about the predictive value of lipoproteins on intracerebral haemorrhage (ICH). Our recent research series showed that higher non-high-density lipoprotein cholesterol (non-HDLC) was an independent predictor of favourable 3-month outcome in ICH patients, we thus aimed to further investigate the association between non-HDLC levels and 1-year functional outcomes after ICH.

**Design** Prospective multicentre cohort study.

**Setting** 13 hospitals in Beijing, China.

**Participants** A total of 666 ICH patients were included between December 2014 and September 2016.

**Methods** Non-HDLC was calculated by subtracting HDL-C from total cholesterol. Patients were then grouped by non-HDLC levels into three categories: <3.4 mmol/L, 3.4–4.2 mmol/L and ≥4.2 mmol/L. Both the univariate and multivariate logistic regressions were used to assess the association between non-HDLC levels and 1-year unfavourable functional outcomes (modified Rankin Scale ≥3) in ICH patients. Moreover, sensitivity analysis was performed in ICH patients without statin use after admission.

**Results** There were 33.5% (223/666) ICH patients identified with unfavourable functional outcomes at 1-year follow-up. In the univariate analysis, patients who achieved non-HDLC levels above 4.2 mmol/L had a 49% decreased risk of 1-year poor prognosis (OR 0.51, 95% CI 0.33 to 0.81). However, non-HDLC did not retain its independent prognostic value in multivariate analysis, the fully adjusted OR values were 1.00 (reference), 1.06 (0.63, 1.79) and 0.83 (0.45, 1.54) from the lowest to the highest non-HDLC group. Moreover, statin use after ICH onset made no difference to the long-term prognosis.

**Conclusions** Non-HDLC was not an independent predictor for 1-year functional outcome in ICH patients, irrespective of poststroke statin use. The predictive value of well-recognised confounding factors was more dominant than non-HDLC on long-term prognosis.

## INTRODUCTION

Intracerebral haemorrhage (ICH) is the second most common subtype of stroke, leading to severe disability and mortality.[1]

## STRENGTHS AND LIMITATIONS OF THIS STUDY

⇒ A multicentre, prospective, cohort study included 666 intracerebral haemorrhage (ICH) patients from a total of 13 hospitals in Beijing.

⇒ Our study filled the vacancy about the association between non-high-density lipoprotein cholesterol (HDLC) and 1-year functional outcomes, simultaneously shed light on the diverse impacts of non-HDLC on short-term and long-term prognosis in ICH patients.

⇒ Sensitivity analysis was performed to evaluate the association between non-HDLC and 1-year functional outcomes in ICH patients with poststroke statin use.

⇒ Data regarding haematoma expansion and antithrombotic treatment were unavailable, further exploration is needed to verify our results.

Based on the nationally representative stroke survey in China published recently, ICH accounts for 25% of all strokes with an overall age-standardised incidence of 66.2 per 100 000 person-years.[2] Despite rapid advances in medicine, the management of ICH remains supportive without significant breakthroughs.[3] Approximately 30%–48% of ICH patients died within 1 month in low-income to middle-income countries and only 12%–39% of survivors could achieve long-term functional independence.[1 4]

The conventional view on lipid-lowering targets goes 'the lower, the better' in patients with atherosclerotic cardiovascular disease. However, previous epidemiology studies suggested an inverse association between lipoprotein cholesterols and ICH risk, haematoma expansion and mortality.[5 6] Much remains to be discussed on the predictive value of lipoproteins on ICH. Our recent research series showed that low serum lipid

levels were independent predictors of 3-month poor prognosis in ICH patients, and non-high-density lipoprotein cholesterol (non-HDLC) was the optimal parameter with high specificity.[7 8] However, the literature has scant information regarding the association between non-HDLC and long-term ICH prognosis.

We, thus, aimed to investigate the association between serum non-HDLC levels and 1-year functional outcomes after ICH in this prospective cohort study.

## METHODS
### Study population
Our study is a multicentre, prospective, cohort study conducted in a total of 13 hospitals, evaluating the medical quality of cerebral haemorrhage on different etiologies in Beijing. From December 2014 to September 2016, 1964 consecutive ICH patients agreed to participate in the study. A total of 1881 patients met the following inclusion criteria: (1) aged 18 years or older and (2) had their first CT scan done within 72 hours after symptom onset. After excluding 159 secondary ICH patients (caused by trauma, tumour, aneurysm, arteriovenous malformation, coagulopathy or other causes) and 20 patients diagnosed as primary ventricular haemorrhage, 1702 patients with primary intraparenchymal haemorrhage were included. Moreover, 294 patients underwent surgical procedures (including craniotomy hematoma removal, haematoma puncture, extraventricular drainage and so on), 15 patients with anticoagulant therapy before symptom onset, 588 patients with missing data on the non-HDLC level and 139 patients lost to follow-up at 1 year were excluded. Eventually, 666 patients with spontaneous ICH from 13 sites were included (figure 1).

### Baseline information
Demographic information including age, sex, onset to admission time, a medical history (including hypertension, diabetes mellitus, hyperlipidaemia, cerebral

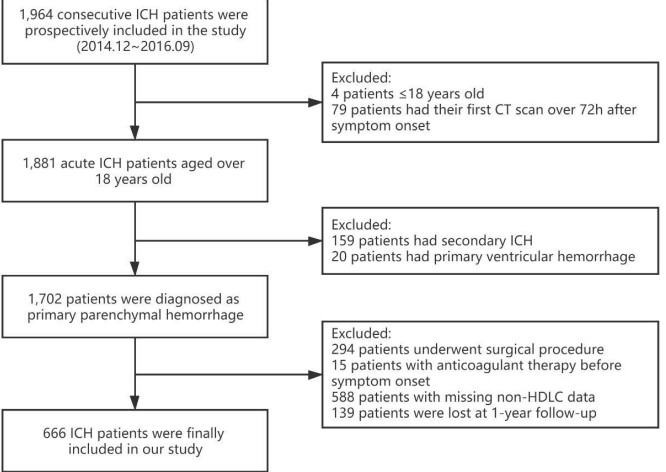

**Figure 1** Flow chart for selection of study participants. ICH, intracerebral haemorrhage; non-HDLC, non-high-density lipoprotein cholesterol.

infarction and ICH), personal habits (including smoking and drinking status) and medication history (including antiplatelet and statin therapy) of each patient was collected using a standard questionnaire at baseline. Neurological deficits were assessed using the National Institute of Health Stroke Scale (NIHSS) and Glasgow Coma Scale (GCS) score by experienced neurologists on admission. Meanwhile, systolic and diastolic blood pressure (BP) (SBP) were measured. A cranial CT scan was performed on admission and haematoma volume was then calculated as ABC/2 volumetric formula at each site.[9] The location of haematoma was further subdivided into supratentorial and infratentorial regions. ICH score was calculated based on five parameters, GCS score, ICH volume, the presence of intraventricular extension, location of haematoma and age.[10]

### Measurement of non-HDLC levels and other biochemical parameters
Blood samples were drawn from the antecubital vein the next morning after an overnight fast and analysed within 4 hours. Total cholesterol (TC) was measured using the end-point test method and HDL-C was measured using a direct method. Non-HDLC was thus calculated by subtracting HDL-C from TC. Based on the National Lipid Association Recommendations,[11] non-HDLC levels were categorised into five groups: desirable, <3.4 mmol/L; above desirable, 3.4–4.2 mmol/L; borderline high, 4.2–5.0 mmol/L; high, 5.0–5.8 mmol/L; and very high, ≥5.8 mmol/L. Accordingly, we integrated the last three groups into one group (≥4.2 mmol/L) due to the limited number of patients.

For other biochemical parameters, random blood glucose was measured via the hexokinase/glucose-6-phosphate dehydrogenase method, serum creatinine was measured through rate reflectance spectrophotometry, white cell count (WCC) together with platelet count were performed on EDTA with an ADVIA 120 counter (Siemens Healthcare Diagnostics, Saint-Denis, France).

### Follow-up information and definition of 1-year ICH prognosis
Patients were followed up at 1 year after ICH onset via telephone interviews. Follow-up evaluation was performed by neurologists who were blinded to prognostic factors. A 1-year prognosis of patients was evaluated by modified Ranking Scale (mRS) score and categorised as favourable (mRS <3) and unfavourable functional outcome groups (mRS ≥3). Newly diagnosed stroke and the subtypes of stroke (both ischaemic stroke and ICH) during the 1-year follow-up period were also documented.

### Patient and public involvement
No patients were involved.

### Statistical analysis
The patients were divided into three groups according to the clinical diagnosis of abnormal non-HDLC levels. Continuous variables were presented as median with IQR, categorical variables were described as count with

percentage. The group differences of continuous variables were analysed using analysis of variance or Kruskal-Wallis test as appropriate, and for categorical variables, $\chi^2$ tests were performed. Logistic regression was used to evaluate the association between non-HDLC levels and 1-year prognosis of ICH patients, with the lowest non-HDLC group (<3.4 mmol/L) used as the reference. Both the univariate and multivariate analyses were conducted to estimate the ORs and 95% CIs. Kaplan-Meier curves were generated and the log-rank test was employed to perform comparisons between the non-HDLC levels. Cox proportional hazards regression analysis was used to evaluate the risk of stroke and stroke subtypes, expressed as the HRs and 95% CIs. Multiple regression models were run as follows. Model 1 was adjusted for age and sex. Model 2 was adjusted for variates in model 1 plus premorbid mRS score (<3 or ≥3), history of ICH, glucose on admission, WCC on admission, baseline haematoma volume, haematoma location, time from onset to initial non-contrast CT (NCCT), GCS score at admission and SBP. P values for trend were conducted using the three categories of non-HDLC as ordinal variables in the model. n addition, sensitivity analysis was performed in ICH patients without statin use after admission (n=589). A two-sided value of p<0.05 was considered statistically significant. All statistical analyses were performed using SAS software, V.9.4 (SAS Institute).

## RESULTS

A total of 666 eligible patients were included, with a mean age of 59 years old (ranging from 51 to 68) and 69.1% (460/666) of them were males. Among them, 33.5% (223/666) were identified as 1-year poor outcomes, the proportion of which were 38.4%, 30.3% and 24.2% from <3.4 mmol/L group to ≥4.2 mmol/L group.

### Baseline characteristics

There were significant differences in age, prior statin use, diastolic BP, glucose on admission, WCC on admission and statin use after admission among the three categories of non-HDLC levels (p<0.05, table 1). Those with higher lipid levels were more likely to be younger, not a prior statin user, having higher diastolic BP and glucose on admission. While no statistical significance was observed in sex, premorbid mRS scale, onset to admission time, past medical history, personal habits, prior antiplatelet use, NIHSS score, GCS score, SBP, creatinine, infections, time from onset to initial NCCT, haematoma volume, haematoma location and ICH score between the three groups.

### Correlation between baseline non-HDLC and 1-year prognosis in ICH patients

In the univariate analysis, higher non-HDLC levels were significantly associated with decreased risk of 1-year poor outcome (p=0.002). Patients who achieved non-HDLC above 4.2 mmol/L had a 49% lower risk of poor

functional outcome at 1 year (OR 0.51, 95% CI 0.33 to 0.81). While no statistical difference was retained after adjusting for age, sex and potential confounding factors (p>0.05). In the fully adjusted model (model 2), the OR values were 1.00 (reference), 1.06 (0.63, 1.79) and 0.83 (0.45, 1.54) from the lowest to the highest non-HDLC group. Moreover, the results maintained consistency in sensitivity analysis among patients without statin use after admission (p=0.842, table 2).

Notably, age, premorbid mRS score (<3 or ≥3) and baseline haematoma volume were positively associated with 1-year poor prognosis in the multivariate analysis. Whereas, higher GCS score at admission was an independent predictor of favourable outcomes. Additional detailed information was given in figure 2.

In the process of statistics, we also calculated the association between the quartiles of non-HDLC with 1-year poor outcome (data were shown in online supplemental table 1). The highest quartile of non-HDLC was significantly associated with decreased risk of 1-year poor outcome, while no statistical difference was retained after adjusting for confounding factors. Due to the identical results of the two cut-off methods, we thus chose the risk-stratified levels of non-HDLC, which had more instructive clinical significance.

### Correlation between baseline non-HDLC and stroke risk

We further investigated the correlation between non-HDLC levels and another stroke (ischaemic or haemorrhagic) risk. In univariate analysis, the cumulative incidences of total stroke, ischaemic stroke and ICH were not statistically different among non-HDLC levels (log-rank test, all p>0.05, figure 3). In multivariate analysis, no correlation was identified between the three groups either (table 3). When the quartile of non-HDLC was set as the cut-off, similar negative results were also obtained (data were not shown).

## DISCUSSION

This study provided evidence on the association between non-HDLC levels and long-term functional outcomes in ICH patients. Although non-HDLC was a significant 1-year predictor in univariate analysis, it did not retain its independent prognostic value in multivariate analysis. Moreover, statin use after ICH onset made no difference to the long-term prognosis.

In our study, the prevalence of 1-year functional independence in ICH patients was 66.5% (443/666), far outweighing the data previously reported.[4] According to the inclusion and exclusion criteria, severe cases who underwent surgical treatment or lost to follow-up were not enrolled. It is noteworthy that per 1 mmol/L increment in non-HDLC yielded a 29% decreased risk of 1-year poor prognosis (crude OR 0.71, 95% CI 0.58 to 0.88). However, contrary to our previous research finding of the independent role of non-HDLC on short-term functional outcomes,[7] the results of this study showed

Table 1  Baseline characteristics of participants according to non-HDLC levels

| | | Non-HDLC levels | | | |
| | Total | <3.4 mmol/L | 3.4–4.2 mmol/L | ≥4.2 mmol/L | P value |
|---|---|---|---|---|---|
| n (%) | 666 | 359 (53.9) | 175 (26.3) | 132 (19.8) | |
| Age, years | 59 (51, 68) | 61 (53, 70) | 57 (49, 67) | 54 (48, 64) | <0.001 |
| Male, n (%) | 460 (69.1) | 258 (71.9) | 120 (68.6) | 82 (62.1) | 0.116 |
| Onset to admission time, h | 4.0 (1.8, 11.9) | 3.8 (1.7, 11.1) | 4.0 (2.0, 11.0) | 4.0 (1.8, 14.7) | 0.840 |
| Premorbid mRS score | | | | | 0.614 |
| mRS <3 | 643 (96.5) | 345 (96.1) | 171 (97.7) | 127 (96.2) | |
| mRS ≥3 | 23 (3.5) | 14 (3.9) | 4 (2.3) | 5 (3.8) | |
| Hypertension, n (%) | 479 (71.9) | 256 (71.3) | 124 (70.9) | 99 (75.0) | 0.676 |
| Diabetes mellitus, n (%) | 106 (15.9) | 55 (15.3) | 29 (16.6) | 22 (16.7) | 0.902 |
| Hyperlipidaemia, n (%) | 68 (10.2) | 36 (10.0) | 18 (10.3) | 14 (10.6) | 0.982 |
| History of CI, n (%) | 102 (15.3) | 58 (16.2) | 27 (15.4) | 17 (12.9) | 0.670 |
| History of ICH, n (%) | 20 (3.0) | 15 (4.2) | 3 (1.7) | 2 (1.5) | 0.141 |
| Smoking, n (%) | 223 (33.5) | 127 (35.4) | 57 (32.6) | 39 (29.6) | 0.458 |
| Drinking, n (%) | 256 (38.4) | 139 (38.7) | 69 (39.4) | 48 (36.4) | 0.850 |
| Prior antiplatelet use, n (%) | 110 (16.5) | 61 (17.0) | 28 (16.0) | 21 (15.9) | 0.771 |
| Prior statin use, n (%) | 44 (6.6) | 31 (8.6) | 10 (5.7) | 3 (2.3) | 0.036 |
| NIHSS score on admission | 8 (3, 13) | 9 (3, 15) | 7 (3, 13) | 5 (2, 12) | 0.083 |
| GCS score on admission | 14 (12, 15) | 14 (12, 15) | 15 (13, 15) | 15 (13, 15) | 0.063 |
| SBP on admission, mm Hg | 160 (149, 183) | 160 (150, 180) | 160 (145, 183) | 162 (150, 183) | 0.564 |
| DBP on admission, mm Hg | 95 (83, 105) | 92 (80, 102) | 96 (85, 106) | 97 (85, 109) | 0.024 |
| Glucose on admission, mmol/L | 6.9 (5.9, 8.4) | 6.6 (5.8, 8.1) | 7.0 (5.9, 8.6) | 7.1 (6.0, 9.3) | 0.032 |
| WCC on admission, ×$10^9$/L | 8.4 (6.6, 10.9) | 8.1 (6.3, 10.7) | 9.1 (7.0, 11.7) | 7.1 (6.0, 9.3) | 0.007 |
| Platelets on admission, ×$10^9$/L | 212 (175, 252) | 202 (164, 238) | 218 (180, 259) | 230 (192, 265) | <0.001 |
| Creatinine on admission, μmol/L | 64.0 (53.0, 77.3) | 64.6 (54.0, 76.4) | 65.0 (52.3, 79.0) | 62.0 (50.1, 76.0) | 0.223 |
| Statin use after admission, n (%) | 77 (11.6) | 19 (5.3) | 30 (17.1) | 28 (21.2) | <0.001 |
| Infections, n (%) | 136 (20.4) | 77 (21.5) | 39 (22.3) | 20 (15.2) | 0.239 |
| Time from onset to initial NCCT, hour | 5.2 (2.3, 16.7) | 5.2 (2.2, 14.8) | 5.1 (2.3, 19.6) | 4.8 (2.3, 19.4) | 0.738 |
| Baseline haematoma volume, mL | 10.5 (5.0, 23.4) | 10.7 (5.0, 25.0) | 10.4 (5.5, 23.1) | 10.0 (4.9, 16.8) | 0.379 |
| Haematoma location | | | | | 0.251 |
| Supratentorial, n (%) | 599 (89.7) | 327 (91.2) | 155 (88.2) | 117 (87.5) | |
| Infratentorial, n (%) | 67 (10.3) | 31 (8.8) | 23 (11.8) | 16 (12.5) | |
| Secondary ventricular haemorrhage | 181 (27.2) | 100 (27.9) | 43 (24.6) | 38 (28.8) | 0.652 |
| ICH score | 0 (0, 1) | 0 (0, 1) | 0 (0, 1) | 0 (0, 1) | 0.447 |

Values are (%) for categorical variables and median (IQR) for continuous variables.
CI, cerebral infarction; DBP, diastolic blood pressure; GCS, Glasgow Coma Scale; ICH, intracerebral haemorrhage; mRS, modified Rankin Scale; NCCT, non-contrast CT; NIHSS, National Institutes of Health Stroke Scale; Non-HDLC, non-high-density lipoprotein cholesterol; SBP, systolic blood pressure; WCC, white cell count.

that age, premorbid mRS score, baseline haematoma volume, admission GCS score, rather than non-HDLC level, were independent predictors for long-term functional outcomes in ICH patients. The validated predictors mentioned above kept high conformity with the items in ICH Functional Outcome Score, an effective prognostic model for 1-year poor functional outcomes after ICH,[12] whereas the absolute magnitude effect of low non-HDLC level on ICH prognosis was likely to be small and overshadowed with time. Beyond that, the amount of rehabilitation with functional gains might also be related.[13]

It was reported that low levels of LDL-C and TC were associated with haematoma expansion.[14 15] As containing all the atherogenic lipoproteins, non-HDLC was served as the preferred target of lipid-lowering therapy.[16] The potential mechanisms regarding the association between

**Table 2** ORs and 95% CI for 1-year poor outcome (mRS ≥3) according to non-HDLC levels

| | Non-HDLC levels | | | Continuous scale | P for trend |
|---|---|---|---|---|---|
| | <3.4 mmol/L | 3.4–4.2 mmol/L | ≥4.2 mmol/L | | |
| 1 year poor outcome, n (%) | 138 (38.4) | 53 (30.3) | 32 (24.2) | | |
| Univariate analysis | Ref. | 0.70 (0.47, 1.02) | 0.51 (0.33, 0.81) | 0.71 (0.58, 0.88) | 0.002 |
| Multivariate analysis | | | | | |
| Model 1 | Ref. | 0.82 (0.54, 1.23) | 0.66 (0.41, 1.06) | 0.81 (0.65, 1.02) | 0.075 |
| Model 2 | Ref. | 1.06 (0.63, 1.79) | 0.83 (0.45, 1.54) | 0.89 (0.76, 1.05) | 0.694 |
| Sensitivity analysis | Ref. | 0.92 (0.53, 1.61) | 1.12 (0.58, 2.16) | 0.92 (0.78, 1.08) | 0.842 |

Data are OR (95% CI) unless otherwise stated.
Model 1 adjusted for age and sex.
Model 2 adjusted for variates in model 1 plus premorbid mRS score (<3 or ≥3), history of ICH, glucose on admission, WCC on admission, baseline haematoma volume, haematoma location, time from onset to initial non-contrast CT, GCS score at admission, systolic blood pressure.
Sensitivity analysis was performed in ICH patients without statin use after admission (n=589), and adjusted for variates in model 2.
GCS, Glasgow Coma Scale; ICH, intracerebral haemorrhage; mRS, modified Ranking Scale; Non-HDLC, non-high-density lipoprotein cholesterol; WCC, white cell count.

hypolipidaemia and haematoma expansion, including impaired endothelial integrity,[17] necrotic medial smooth muscle cells,[18] increased erythrocyte fragility,[19] inhibited platelet aggregation[20] and the resultant incident cerebral microbleeds.[21] Despite the theoretical basis, our study failed to show an independent correlation between non-HDLC levels and 1-year functional outcomes in ICH patients. The secondary injury caused by low levels of lipoproteins in ICH patients was associated with short-term prognosis (30 days and 3 months),[22 23] while its impact on long-term prognosis (1 year) was negative, which merits further investigation due to the limited sample size and incomplete neuroimaging data on haematoma expansion in our study.

Statin treatment is another major concern,[24] there were respectively 6.6% (44/666) and 11.6% (77/666) patients with prestroke and poststroke statin use in our study. Two recent meta-analyses concluded that there was no

evidence to suggest prestroke statin therapy may increase bleeding risk in the context of ICH.[25 26] Whether to start, continue, or stop statin treatment in ICH patients has aroused great concern, we thus conducted a sensitivity analysis to evaluate the effect of statin exposure after admission on ICH prognosis. No significant difference was detected between non-HDLC levels and 1-year prognosis in ICH patients in our study, irrespective of poststroke statin use. A recent review indicated that statin should be applied after weighing the pros and cons given its pleiotropic as well as lipid-lowering effects.[27] Because of the relatively low stain exposure rate in our study, it is necessary to conduct randomised controlled trials around this topic.

Our study filled the vacancy about the association between non-HDLC and 1-year functional outcomes, simultaneously shed light on the diverse impacts of non-HDLC on short-term and long-term prognosis in ICH patients. Nonetheless, there are still some limitations. First, the follow-up radiological information was unavailable, which makes it difficult to verify the intermediate role of haematoma expansion between non-HDLC and poor prognosis. Second, ICH caused by cerebral amyloid angiopathy has a higher rebleeding risk than hypertensive one,[28] while data regarding the cause of ICH was not documented in our study. Despite no correlation was observed between the history of ICH and 1-year functional

| Variables | Poor outcome (mRS ≥3, n=178) | Odds ratio (95% CI) |
|---|---|---|
| non-HDLC (vs <3.4mmol/L) | | |
| 3.4-4.2mmol/L | 1.06 (0.63, 1.79) | |
| ≥4.2mmol/L | 0.83 (0.45, 1.54) | |
| Age, years | 1.08 (1.06, 1.10) | |
| Male | 0.84 (0.52, 1.36) | |
| Premorbid mRS score ≥3 | 3.89 (1.43, 6.77) | |
| History of ICH | 2.80 (0.95, 5.27) | |
| Glucose on admission, mmol/L | 1.02 (0.95, 1.11) | |
| WBC on admission, 10E9/L | 1.00 (0.93, 1.08) | |
| Platelet on admission, 10E9/L | 1.00 (0.99, 1.00) | |
| Baseline hematoma volume, mL | 1.02 (1.01, 1.04) | |
| Infratentorial hemorrhage | 0.82 (0.37, 1.82) | |
| Tme from onset to initial NCCT, hours | 0.99 (0.98, 1.01) | |
| GCS score on admission | 0.73 (0.66, 0.80) | |
| SBP, mmHg | 0.99 (0.99, 1.00) | |

**Figure 2** Multivariate predictors of 1-year poor outcome among ICH patients. GCS, Glasgow Coma Scale; HDLC, high-density lipoprotein cholesterol; ICH, intracerebral haemorrhage; MRS, modified Rankin scale; NCCT, noncontrast CT; SBP, systolic blood pressure; WCC, white cell count.

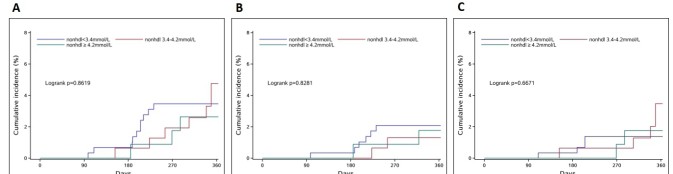

**Figure 3** Cumulative incidences of (A) total stroke, (B) ischaemic stroke and (C) intracerebral haemorrhage according to non-HDLC levels. Non-HDLC, non-high-density lipoprotein cholesterol.

**Table 3** HR for stroke according to non-HDLC levels

| | Non-HDLC levels | | | | |
|---|---|---|---|---|---|
| | <3.4 mmol/L | 3.4–4.2 mmol/L | ≥4.2 mmol/L | P for trend | Per 1 SD increase |
| **Total stroke** | | | | | |
| Events, n (%) | 10 (2.8) | 6 (3.4) | 3 (2.3) | | |
| Model 1 | Ref. | 1.06 (0.38, 2.94) | 0.71 (0.19, 2.61) | 0.88 (0.49, 1.59) | 0.96 (0.67, 1.39) |
| Model 2 | Ref. | 1.44 (0.50, 4.22) | 0.83 (0.21, 3.25) | 0.98 (0.54, 1.80) | 1.00 (0.74, 1.35) |
| **Ischaemic stroke** | | | | | |
| Events, n (%) | 6 (1.7) | 2 (1.1) | 2 (1.5) | | |
| Model 1 | Ref. | 0.56 (0.11, 2.79) | 0.75 (0.15, 3.79) | 0.81 (0.35, 1.86) | 0.94 (0.61, 1.47) |
| Model 2 | Ref. | 0.73 (0.14, 3.89) | 0.65 (0.12, 3.67) | 0.80 (0.34, 1.86) | 0.99 (0.75, 1.32) |
| **Intracerebral haemorrhage** | | | | | |
| Events, n (%) | 4 (1.1) | 4 (2.3) | 2 (1.5) | | |
| Model 1 | Ref. | 1.86 (0.46, 7.52) | 1.24 (0.22, 6.89) | 1.18 (0.55, 2.54) | 1.01 (0.53, 1.94) |
| Model 2 | Ref. | 2.84 (0.61, 13.14) | 1.80 (0.28, 11.53) | 1.41 (0.63, 3.19) | 1.07 (0.52, 2.21) |

Data are HR (95% CI) unless otherwise stated.
Model 1 adjusted for age and sex.
Model 2 adjusted for variates in model 1 plus prior mRS scale (<3 or ≥3) history of ICH, glucose on admission, WCC on admission, baseline haematoma volume, haematoma location, time from onset to initial non-contrast CT, GCS score at admission, systolic blood pressure.
GCS, Glasgow Coma Scale; ICH, intracerebral haemorrhage; mRS, modified Ranking Scale; Non-HDLC, non-high-density lipoprotein cholesterol; WCC, white cell count.

outcome, the impact of ICH aetiology merits further investigation. Third, medication therapy regarding antiplatelet or anticoagulation agents were not included in the multivariate analysis, whereas accumulating researches proved that antithrombotic treatment increased the risk of cerebral microbleeds as well as future ICH.[29 30] Although we collected preictus antiplatelet use, restricted by the small sample size, further research is needed to provide insight into the relationship. Moreover, since our study based on a highly selected population with small haematoma and relatively good neurologic status to achieve precise research, the findings cannot be generalised to the whole ICH population.

## CONCLUSION

In conclusion, non-HDLC was not an independent predictor for 1-year functional outcome in ICH patients, irrespective of poststroke statin use. The predictive value of well-recognised confounding factors was more dominant than non-HDLC on long-term poor prognosis. Further prospective studies are needed to assess the impact of lower non-HLDC levels on ICH prognosis.

**Author affiliations**
[1]Department of Neurology, Beijing Tiantan Hospital, Capital Medical University, Beijing, China
[2]China National Clinical Research Center for Neurological Diseases, Beijing, China
[3]Department of Neurology, Beijing Hospital, National Center of Gerontology, Beijing, China
[4]Research Unit of Artificial Intelligence in Cerebrovascular Disease, Chinese Academy of Medical Sciences, Beijing, China

**Contributors** YW and JW performed the experiments, interpreted the results of statistical analysis, and drafted the manuscript. AW conducted the statistical analysis and interpreted the data. RJ revised the manuscript for intellectual content. WW and XZ had full access to all of the data and take responsibility for the integrity of the data and the accuracy of the data analysis. And XZ was the guarantor of our study.

**Funding** Our study was supported by grants from the Chinese Academy of Medical Sciences Innovation Fund for Medical Sciences (2019-I2M-5–029), Beijing Natural Science Foundation (Z200016), Beijing Municipal Committee of Science and Technology (Z201100005620010), and Ministry of Science and Technology of the People's Republic of China (2018YFC1705003).

**Competing interests** None declared.

**Patient and public involvement** Patients and/or the public were not involved in the design, or conduct, or reporting, or dissemination plans of this research.

**Patient consent for publication** Not applicable.

**Ethics approval** The study was approved by the Central Institutional Review Board of Beijing Tiantan Hospital (KY2014-023-02) and written informed consent was obtained. Participants gave informed consent to participate in the study before taking part.

**Provenance and peer review** Not commissioned; externally peer reviewed.

**Data availability statement** Data are available on reasonable request. Some or all datasets generated during and/or analysed during the current study are not publicly available but are available from the corresponding author on reasonable request.

**ORCID iDs**
Yu Wang http://orcid.org/0000-0002-8636-9540
Xingquan Zhao http://orcid.org/0000-0002-2572-2718

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
