## [Reviewer comments · BMJ Open]

ARTICLE DETAILS

TITLE (PROVISIONAL)	Association between non-HDLC and 1-year prognosis in patients with spontaneous intracerebral hemorrhage: a prospective cohort study from 13 hospitals in Beijing
AUTHORS	Wang, Yu; Wu, Jianwei; Wang, Anxin; Jiang, Ruixuan; Wang, Wenjuan; Zhao, Xingquan

VERSION 1 – REVIEW

REVIEWER	ZK Law Universiti Kebangsaan Malaysia Medical Centre, Department of Medicine
REVIEW RETURNED	14-Feb-2022

GENERAL COMMENTS	A well written and presented prospective observational study. I have several comments: 1) this appears to be a follow-up of "Feng H, Wang X, Wang W, Zhao X. Association Between Non-high-density Lipoprotein Cholesterol and 3-Month Prognosis in Patients With Spontaneous Intracerebral Hemorrhage. Front Neurol 2020;11:920." In the prior study, final number included was 654 whilst this study was 666. Please briefly explain the reasons for discrepancies. 2) In the abstract: "Our recent research series showed that non-high-density lipoprotein cholesterol (non-HDLC) was an independent predictor of 3-month poor prognosis in ICH patients"- kindly stated whether it's "higher" non-HDLC. In the limitation section of abstract: "Factors including radiological information or antithrombotic treatment may affect the results"- it's unclear what this means, kindly be more specific. 3) There are 2 possible reasons, in theory, why the first study Feng 2020 (ref#7) showed lower non-HDL association with worse outcome at day 90 whilst the current study did not at one-year: a) the cut-offs used in both analysis are different- Feng 2020 used quartile while current one used fixed levels. What if the same cut-off levels are used? b) non-HDL did have protective effect early on, but later lead to worse outcome. For example did it increased the risk of another ICH/stroke. Perhaps a Kaplan Meier curve of death after ICH for different cut-offs may help illustrate this, if data on recurrent stroke is not available. 4) history of ICH seems to be the most significant prognostic factor (figure 2). Was this because of a poor pre-morbid mRS before entry into the study? or did patients with previous ICH had CAA-ICH, putting them at risk of another ICH/stroke within the study period?
--

	5) Table 2- pre-morbid mRS and presence of IVH are usually important prognostic factors. If available should be include, if not kindly acknowledge as a limitation.
--	---

REVIEWER	Marc-Alain Babi University of Florida
REVIEW RETURNED	29-Mar-2022

GENERAL COMMENTS	The reviewer completed the checklist but made no further comments.
--

REVIEWER	Joji Inamasu Fujita Health University Hospital, Neurosurgery
REVIEW RETURNED	11-Apr-2022

GENERAL COMMENTS	1. While the authors had reported in other journals about the association between non-HDLC levels and poor outcomes at 3 months, the authors fail to explain why that association disappeared at 1 year-followup in this manuscript. It can be assumed that some patients who had been graded as \geqmRS3 at 3 months might improve to <mRS3 after rigorous rehabilitation, which might explain the loss of association at 1 year. 2. While the authors claim that the study design was prospective, it should be noted that the frequency of cases with missing non-HDLC data was quite high (588/1702=34.5% of ICH patients data missing), raising question about the data integrity. Similarly, the frequency of lost-to-followup was not low (139/1702=8.2% of ICH patients were lost to followup at one year). This also raises question about the data integrity. Furthermore, because of exclusion of surgical cases (with large hematoma), only cases with a small hematoma with relatively good neurologic status were included in this study. In other words, it needs to be mentioned that the patient cohort in this manuscript was highly selected, which might have affected the results. 3. While evaluation by phone interview was a simple and convenient method, its shortcoming will be evident the longer the observation period. Particularly the lost-to-followup rate will increase as the followup period is extended. The longer the follow-up period is, the more recurrence of ICH may become problematic. It has been reported that 2.7% of ICH patients experienced recurrence with a mean interval of 13.1 months (Recurrent intracerebral hemorrhage due to hypertension, Neurosurgery 1990). Phone interview to patients may not be able to catch patients with unexpectedly poor neurologic status, such as those with ICH recurrence, 4. ICH Score appeared in Table 1. However, those results were not explained in the text, and no references regarding the ICH Score were cited in the References.
--

VERSION 1 – AUTHOR RESPONSE

Reviewer #1:

1. Comment: This appears to be a follow-up of "Feng H, Wang X, Wang W, Zhao X.

Association Between Non-high-density Lipoprotein Cholesterol and 3-Month Prognosis in Patients With Spontaneous Intracerebral Hemorrhage. Front Neurol 2020;11:920." In the prior study, final number included was 654 whilst this study was 666. Please briefly explain the reasons for discrepancies.

Response: Thanks for your careful review. The discrepancies in the numbers of enrolled patients between our present study and the prior research lied in the exclusion of patients missing baseline hematoma volume and the number of patients lost to follow-up. Our present study replaced the missing hematoma volume with the median value, while the prior study excluded the 19 patients with missing data on hematoma volume. Additionally, there were 139 patients failed in the 1-year telephone interview, compared with 132 patients in the 3-month follow-up. For the above-mentioned reasons, the final number in our study was 666 whilst the prior study was 654.

- 2. Comment: In the abstract: "Our recent research series showed that non-high-density lipoprotein cholesterol (non- HDLC) was an independent predictor of 3-month poor prognosis in ICH patients"- kindly stated whether it's "higher" non-HDLC. In the limitation section of abstract: "Factors including radiological information or antithrombotic treatment may affect the results"- it's unclear what this means, kindly be more specific.**

Response: Thanks for your careful review. We apologize for the vaguely-stated part of our study and agreed that the descriptions should be more specific. The second sentence in the Objectives part of Abstract was revised as (Page 2, Line 4-6) "Our recent research series showed that higher non-high-density lipoprotein cholesterol (non-HDLC) was an independent predictor of favourable 3-month outcome in ICH patients". In the limitation section of Abstract, the sentence was rewritten as (Page 3, Line 13-14) "Data regarding radiological information and antithrombotic treatment were unavailable, further exploration is needed to verify our results".

- 3. Comment: There are 2 possible reasons, in theory, why the first study Feng 2020 (ref#7) showed lower non-HDL association with worse outcome at day 90 whilst the current study**

did not at one-year: a) the cut-offs used in both analysis are different- Feng 2020 used quartile while current one used fixed levels. What if the same cut-off levels are used? b) non-HDL did have protective effect early on, but later lead to worse outcome. For example did it increased the risk of another ICH/stroke. Perhaps a Kaplan Meier curve of death after ICH for different cut-offs may help illustrate this, if data on recurrent stroke is not available.

Response: Thanks for your careful review. In the process of statistics, we calculated the association between both the risk-stratified levels and quartiles of non-HDLC with 1-year poor outcome (results of non-HDLC quartiles were shown in Table 1 below). The highest quartile of non-HDLC was significantly associated with decreased risk of 1-year poor outcome, while no statistical difference was retained after adjusting for confounding factors. Due to the identical results of the two cut-off methods, we thus chose the risk-stratified levels of non-HDLC as it had more instructive clinical significance.

Table 1. Odds ratios and 95% CI for 1-year poor outcome (mRS ≥ 3) according to non-HDLC quartiles.

	non-HDLC quartiles				Continuous scale	P for trend
	Q1	Q2	Q3	Q4		
1-year poor outcome, n (%)	71 (43.3)	58 (34.5)	54 (32.3)	40 (24.0)		
Univariate analysis	Ref.	0.69 (0.44, 1.08)	0.63 (0.40, 0.98)	0.41 (0.26, 0.66)	0.76 (0.66, 0.88)	<0.001
Multivariate analysis						
Model 1	Ref.	0.80 (0.50, 1.29)	0.84 (0.52, 1.36)	0.57 (0.35, 0.95)	0.85 (0.73, 1.00)	0.049
Model 2	Ref.	0.81 (0.44, 1.50)	1.03 (0.56, 1.90)	0.71 (0.37, 1.37)	0.93 (0.76, 1.14)	0.468
Sensitivity analysis	Ref.	0.83 (0.43, 1.60)	1.14 (0.60, 2.18)	0.76 (0.39, 1.51)	0.96 (0.77, 1.18)	0.673

Data are OR (95% CI) unless otherwise stated.

Model 1 adjusted for age and sex.

Model 2 adjusted for variates in model 1 plus prior mRS scale (<3 or ≥3) history of ICH, glucose on admission, WBC on admission, baseline hematoma volume, hematoma location, time from onset to initial non-contrast CT, GCS score at admission, systolic blood pressure.

Sensitivity analysis was performed in ICH patients without statin use after admission (n=589), and adjusted for variates in model 2.

We further investigated the correlation between non-HDL-C levels and another stroke (ischemic or hemorrhagic) risk upon the suggestion. In univariate analysis, the cumulative incidences of total stroke, IS, and ICH were not statistically different among non-HDL-C levels (log-rank test, all $P > 0.05$, Figure 1). In multivariate analysis, no correlation was identified between the three groups either (Table 2). When the quartile of non-HDL-C was set as the cut-off, similar negative results were also obtained (data was not shown).

Figure 1. Cumulative incidences of (A) total stroke, (B) ischemic stroke, and (C) intracerebral hemorrhage according to non-HDL-C levels.

Table 2. Hazard ratios (HR) for stroke according to non-HDL-C levels.

	non-HDL-C levels			P for trend	Per 1 SD increase
	<3.4mmol/L	3.4-4.2mmol/L	≥4.2mmol/L		
Total stroke					
Events, n (%)	10 (2.8)	6 (3.4)	3 (2.3)		
Model 1	Ref.	1.06 (0.38, 2.94)	0.71 (0.19, 2.61)	0.88 (0.49, 1.59)	0.96 (0.67, 1.39)
Model 2	Ref.	1.44 (0.50, 4.22)	0.83 (0.21, 3.25)	0.98 (0.54, 1.80)	1.00 (0.74, 1.35)

Ischemic stroke					
Events, n (%)		6 (1.7)	2 (1.1)	2 (1.5)	
Model 1	Ref.	0.56 (0.11, 2.79)	0.75 (0.15, 3.79)	0.81 (0.35, 1.86)	0.94 (0.61, 1.47)
Model 2	Ref.	0.73 (0.14, 3.89)	0.65 (0.12, 3.67)	0.80 (0.34, 1.86)	0.99 (0.75, 1.32)
Intracerebral hemorrhage					
Events, n (%)		4 (1.1)	4 (2.3)	2 (1.5)	
Model 1	Ref.	1.86 (0.46, 7.52)	1.24 (0.22, 6.89)	1.18 (0.55, 2.54)	1.01 (0.53, 1.94)
Model 2	Ref.	2.84 (0.61, 13.14)	1.80 (0.28, 11.53)	1.41 (0.63, 3.19)	1.07 (0.52, 2.21)

Data are HR (95% CI) unless otherwise stated.

Model 1 adjusted for age and sex.

Model 2 adjusted for variates in model 1 plus prior mRS scale (<3 or ≥3) history of ICH, glucose on admission, WBC on admission, baseline hematoma volume, hematoma location, time from onset to initial non-contrast CT, GCS score at admission, systolic blood pressure.

4. Comment: History of ICH seems to be the most significant prognostic factor (figure 2). Was this because of a poor premorbid mRS before entry into the study? or did patients with previous ICH had CAA-ICH, putting them at risk of another ICH/stroke within the study period?

Response: We appreciate your insightful suggestion. The percentage of poor premorbid mRS was 3.5% (23/666) in the total patients. And there was 3% (20/666) of patients had a history of ICH, for whom 5.0% (1/20) had a poor premorbid mRS. To verify the prognostic value of premorbid mRS, we thus added the factor -- premorbid mRS score (<3 or ≥3) into the multivariate model 2. The OR value of premorbid mRS score was 3.89 (1.43, 6.77), while the significance of the history of ICH diminished indeed (OR 2.80, 95% CI 0.95-5.27). We thus revised the main manuscript and updated Figure 2 accordingly.

While the cause of previous ICH (hypertensive or CAA) was not collected in our study. We recognized the limitation and mentioned it in the last paragraph of the Discussion part (Page 13, Line 1-5): “ Secondly, ICH caused by cerebral amyloid angiopathy has a higher rebleeding risk than hypertensive one (*Neurology*. 2018; 90(2): e119-e126), while data regarding the cause of

ICH was not documented in our study. Despite no correlation was observed between the history of ICH and 1-year functional outcome, the impact of ICH etiology merits further investigation”.

Figure 2. Multivariate predictors of 1-year poor outcome among ICH patients. Non-HDLC, non-high-density lipoprotein cholesterol; ICH, intracerebral hemorrhage; mRS, modified Rankin Scale; WBC, white blood cells; NCCT, non-contrast CT; GCS, Glasgow Coma Scale; SBP, systolic blood pressure.

5. Comment: Table 2- premorbid mRS and presence of IVH are usually important prognostic factors. If available should be include, if not kindly acknowledge as a limitation.

Response: Thanks for your careful review. Premorbid mRS (dichotomized into <3 and \geq 3) was added into the multivariate model 2, and the result showed that premorbid mRS was independently associated with 1-year functional outcome (OR 3.89, 95% CI 1.43- 6.77). We thus revised the main manuscript and updated Figure 2 accordingly.

Concerning the parameter – the presence of IVH, 20 patients diagnosed with primary ventricular hemorrhage was eliminated in the selection process. We thus did not mention this variable in our manuscript.

Reviewer #2:

Response: Thanks for your review. Previous studies suggested an inverse association between lipoprotein cholesterol and bleeding risk, while limited data was available about the predictive value of lipoproteins on ICH. Our recent research series showed that non-HDL-C was an independent predictor of 3-month poor prognosis in ICH patients, we thus aimed to further investigate the association between non-HDL-C levels and 1-year functional outcomes after ICH in this prospective cohort study. Our study demonstrated that non-HDL-C was not an independent predictor for 1-year functional outcome in ICH patients, irrespective of post-stroke statin use. The predictive value of well-recognized confounding factors was more dominant than non-HDL-C in the long-term prognosis of ICH patients.

Reviewer #3:

- 1. Comment: While the authors had reported in other journals about the association between non-HDL-C levels and poor outcomes at 3 months, the authors fail to explain why that association disappeared at 1 year-follow up in this manuscript. It can be assumed that some patients who had been graded as \geq mRS 3 at 3 months might improve to $<$ mRS 3 after rigorous rehabilitation, which might explain the loss of association at 1 year.**

Response: We appreciate your insightful suggestion and agree that rigorous rehabilitation might explain the loss of association between non-HDL-C levels and 1-year functional outcomes. We thus added the following in the Discussion part (Page 11, Line 17-18): "Beyond that, the amount of rehabilitation with functional gains might also related (*J Stroke Cerebrovasc Dis.* 2019; 28(9): 2537-2542)".

- 2. Comment: While the authors claim that the study design was prospective, it should be noted that the frequency of cases with missing non-HDL-C data was quite high (588/1702=34.5% of ICH patients data missing), raising question about the data integrity. Similarly, the frequency of lost-to-follow up was not low (139/1702=8.2% of ICH patients were lost to followup at one year). This also raises question about the data integrity.**

Furthermore, because of exclusion of surgical cases (with large hematoma), only cases with a small hematoma with relatively good neurologic status were included in this study. In other words, it needs to be mentioned that the patient cohort in this manuscript was highly selected, which might have affected the results.

Response: Thanks for your careful review. We recognized the limitation and compared the baseline characteristics between the included and excluded participants in Table 3. There were significant differences in age, prior antiplatelet use, NIHSS score on admission, GCS on admission, SBP on admission, statin use after admission, infection, WBC, platelet, creatinine on admission, hematoma volume, hematoma location, and ICH score. The included patients were more likely to be older, a prior antiplatelet user, having lower NIHSS score, ICH score, infection rate, hematoma volume, and higher GCS score on admission. The clinical features between the two groups were in accordance with our inclusion and exclusion criteria that patients included had relatively good neurologic status and smaller hematoma. Therefore, we mentioned the limitation in the last paragraph of the Discussion part (Page 12, Line 275-277): “Moreover, since our study based on a highly selected population with small hematoma and relatively good neurologic status to achieve precise research, the findings cannot be generalized to the whole ICH population”.

Table 3. Characteristics of included and excluded ICH participants.

	Total	Included participants	Excluded participants	P -value
n (%)	1964	666 (33.9)	1298 (66.1)	
Age, years	57 (48, 66)	59 (51, 68)	56 (47, 65)	<0.001
Male, n (%)	1327 (67.6)	460 (69.1)	867 (66.8)	0.308
Onset to admission time, h	4.0 (1.9, 11.0)	4.0 (1.8, 11.9)	4.0 (2.0, 10.8)	0.428
Premorbid mRS score				0.870
mRS<3	1898 (96.6)	643 (96.6)	1255 (96.7)	
mRS≥3	66 (3.4)	23 (3.4)	43 (3.3)	
Hypertension, n (%)	1367 (70.1)	479 (71.9)	888 (69.2)	0.215

Diabetes mellitus, n (%)	289 (14.7)	106 (15.9)	183 (14.1)	0.282
Hyperlipidemia, n (%)	184 (9.4)	68 (10.2)	116 (8.9)	0.359
History of CI, n (%)	267 (13.6)	102 (15.3)	165 (12.7)	0.111
History of ICH, n (%)	50 (2.6)	20 (3.0)	30 (2.3)	0.357
Smoking, n (%)	628 (32.0)	223 (33.5)	405 (31.2)	0.305
Drinking, n (%)	716 (36.5)	256 (38.4)	460 (35.4)	0.191
Prior antiplatelet use, n (%)	277 (14.1)	110 (16.5)	167 (12.9)	0.002
Prior statin use, n (%)	105 (5.3)	44 (6.6)	61 (4.7)	0.075
NIHSS score on admission	12 (4, 23)	8 (3, 13)	15 (4, 26)	<0.001
GCS score on admission	14 (8, 15)	14 (12, 15)	12 (6, 15)	<0.001
SBP on admission, mmHg	165 (147, 186)	160 (149, 183)	168 (146, 189)	0.031
DBP on admission, mmHg	96 (82, 109)	95 (83, 105)	96 (82, 110)	0.144
Glucose on admission, mmol/L	7.3 (6.1, 9.2)	6.9 (5.9, 8.4)	7.6 (6.3, 9.7)	<0.001
WBC on admission, 10 ⁹ /L	9.8 (7.4, 13.0)	8.4 (6.6, 10.9)	10.7 (7.8, 14.0)	<0.001
Platelets on admission, 10 ⁹ /L	216 (177, 257)	212 (175, 252)	218 (180, 258)	0.021
Creatinine on admission, µmol/L	63.4 (52.7, 77.0)	64.0 (52.0, 77.3)	63.1 (52.6, 76.6)	0.371
Statin use after admission, n (%)	120 (6.1)	77 (11.6)	43 (3.3)	<0.001
Infections, n (%)	595 (30.3)	136 (20.4)	459 (35.4)	<0.001
Time from onset to initial NCCT, h	4.8 (2.3, 16.7)	5.2 (2.3, 16.7)	4.7 (2.3, 16.7)	0.460
Baseline hematoma volume, ml	15.8 (6.0, 38.6)	10.5 (5.0, 23.4)	20.3 (7.6, 50.3)	<0.001
Hematoma location				<0.001
Supratentorial, n (%)	1780 (90.6)	599 (89.7)	1181 (91.0)	
Infratentorial, n (%)	188 (9.4)	67 (10.3)	121 (9.0)	
ICH score	1 (0, 2)	0 (0, 1)	1 (0, 2)	<0.001

Values are (%) for categorical variables and median (IQR) for continuous variables.

mRS, modified Rankin Scale; CI, cerebral infarction; ICH, intracerebral hemorrhage; NIHSS, National Institutes of Health Stroke Scale; GCS, Glasgow Coma Scale; SBP, systolic blood pressure; DBP, diastolic blood pressure; WBC, white blood cells; NCCT, non-contrast CT.

- 3. Comment: While evaluation by phone interview was a simple and convenient method, its shortcoming will be evident the longer the observation period. Particularly the lost-to-followup rate will increase as the followup period is extended. The longer the follow-up period is, the more recurrence of ICH may become problematic. It has been reported that 2.7% of ICH patients experienced recurrence with a mean interval of 13.1 months (Recurrent intracerebral hemorrhage due to hypertension, Neurosurgery 1990). Phone interview to patients may not be able to catch patients with unexpectedly poor neurologic status, such as those with ICH recurrence.**

Response: Thank you for your comment. Our follow-up process complied with the standard protocol strictly. Patients were contacted via telephone by trained research coordinators 3 and 12 months after enrolment. For patients who were not reached at the first telephone follow-up interview, we conducted follow-up telephone interviews once a week for 3 weeks. If none of the four telephone follow-up interviews were successful, the follow-up was considered as lost. Moreover, the confirmation of cerebrovascular events (including recurrent ICH) were sought from the treating hospital, and suspected recurrent cerebrovascular events without hospitalization were judged by an independent endpoint judgment committee.

- 4. Comment: ICH Score appeared in Table 1. However, those results were not explained in the text, and no references regarding the ICH Score were cited in the References.**

Response: Thanks for your careful review. We apologize for the lack of elaboration on the ICH score in our text. To address your comments, we added the definition of ICH score in the section Methods (Page 6, Line 5-6) "ICH score was calculated based on five parameters, GCS score, ICH volume, the presence of intraventricular extension, location of hematoma, and age (*Stroke*. 2001; 32(4): 891-897)". Moreover, we added the following in the section Results (Page 8, Line 17-21) "While no statistical significance was observed in sex, onset to admission time, past medical history, personal habits, prior antiplatelet use, NIHSS score, GCS score, SBP, creatinine,

infections, time from onset to initial NCCT, hematoma volume, hematoma location, and ICH score between the three groups” to interpret the negative results of Table 1.

Thank you for your help and suggestions. We appreciate all your efforts in reviewing our manuscript during this challenging time. Your generous suggestions have helped us to present our study clearer and more comprehensively.

VERSION 2 – REVIEW

REVIEWER	ZK Law Universiti Kebangsaan Malaysia Medical Centre, Department of Medicine
REVIEW RETURNED	28-Jul-2022

GENERAL COMMENTS	Thank you for your revised manuscript. I have some comments/suggestions as follows: 1) "Data regarding radiological information and antithrombotic treatment were unavailable, further exploration is needed to verify our results." but that's not quite true. You did do haematoma volume measurement and location. perhaps the authors should specify it's haematoma expansion. 2) thank you for reply regarding non-HDL cut-offs "In the process of statistics, we calculated the association between both the risk-stratified levels and quartiles of non-HDLC with 1-year poor outcome (results of non-HDLC quartiles were shown in Table 1 below). The highest quartile of non-HDLC was significantly associated with decreased risk of 1-year poor outcome, while no statistical difference was retained after adjusting for confounding factors. Due to the identical results of the two cut-off methods, we thus chose the risk-stratified levels of non-HDLC as it had more instructive clinical significance." Perhaps this explanation can be included in text, the Table 1 in the response letter could be added as a supplemental material. 3. the figure and table 2 in the response letter regarding the risk of IS and ICH cumulatively looks good, perhaps could be included in text. The results section with 2 tables does appear slightly short. 4. Whilst it's reasonable to exclude primary IVH, many intraparenchymal ICH would be complicated by secondary IVH (~40% according to literature). IVH is a significant prognostic factor with mortality of 50-80% therefore should be shown in the baseline table separately. The authors have shown the ICH score, therefore implying this data is available.
--

REVIEWER	Joji Inamasu Fujita Health University Hospital, Neurosurgery
REVIEW RETURNED	24-Jul-2022

GENERAL COMMENTS	The authors were able to respond to most of the queries posed by the reviewers. The revised version will be suitable for publication in BMJ Open.
---

VERSION 2 – AUTHOR RESPONSE

Reviewer #1:

- 6. Comment: "Data regarding radiological information and antithrombotic treatment were unavailable, further exploration is needed to verify our results." But that's not quite true. You did do haematoma volume measurement and location. perhaps the authors should specify it's haematoma expansion.**

Response: Thanks for your careful review. We apologize for the inaccurate statement and agreed that “radiological information” should be revised as “hematoma expansion”. In the limitation section of Abstract, the sentence was rewritten as (Page 3, Line 12-13) “Data regarding hematoma expansion and antithrombotic treatment were unavailable, further exploration is needed to verify our results”.

- 7. Comment: Thank you for reply regarding non-HDL cut-offs "In the process of statistics, we calculated the association between both the risk-stratified levels and quartiles of non-HDLC with 1-year poor outcome (results of non-HDLC quartiles were shown in Table 1 below). The highest quartile of non-HDLC was significantly associated with decreased risk of 1-year poor outcome, while no statistical difference was retained after adjusting for confounding factors. Due to the identical results of the two cut-off methods, we thus chose the risk-stratified levels of non-HDLC as it had more instructive clinical significance." Perhaps this explanation can be included in text, the Table 1 in the response letter could be added as a supplemental material.**

Response: We appreciate your insightful suggestion and agreed that both the explanation and Table 1 in the response letter should be included. We thus added the above sentence in the fifth paragraph of Results (Page 11, Line 4-9) and uploaded the Table 1 in the response letter as Supplementary Table 1.

- 8. Comment: The figure and table 2 in the response letter regarding the risk of IS and ICH cumulatively looks good, perhaps could be included in text. The results section with 2 tables does appear slightly short.**

Response: Thank you for your comment. We added a section “correlation between baseline non-HDLC and stroke risk” in Results (Page 11, Line 11-17), as “We further investigated the correlation between non-HDLC levels and another stroke (ischemic or hemorrhagic) risk. In univariate analysis, the cumulative incidences of total stroke, ischemic stroke, and ICH were not statistically different among non-HDLC levels (log-rank test, all $P > 0.05$, Figure 3). In multivariate analysis, no correlation was identified between the three groups either (Table 3). When the quartile of non-HDLC was set as the cut-off, similar negative results were also obtained (data was not shown).”

- 9. Comment: Whilst it's reasonable to exclude primary IVH, many intraparenchymal ICH would be complicated by secondary IVH (~40% according to literature). IVH is a significant prognostic factor with mortality of 50-80% therefore should be shown in the baseline table separately. The authors have shown the ICH score, therefore implying this data is available.**

Response: We appreciate your insightful suggestion and added the variable “secondary ventricular hemorrhage” into Table 1. Among the 666 included patients, 27.2% (181/666) had secondary ventricular hemorrhage, the proportion of which were 27.9%, 24.6%, and 28.8% from <3.4mmol/L group to ≥ 4.2 mmol/L group.

Reviewer #3:

- 5. Comment: The authors were able to respond to most of the queries posed by the reviewers. The revised version will be suitable for publication in BMJ Open.**

Response: We appreciate all your efforts in reviewing our manuscript.

Thank you for your help and suggestions. We appreciate all your efforts in reviewing our manuscript during this challenging time. Your generous suggestions have helped us to present our study clearer and more comprehensively.

VERSION 3 – REVIEW

REVIEWER	ZK Law Universiti Kebangsaan Malaysia Medical Centre, Department of Medicine
REVIEW RETURNED	17-Sep-2022
GENERAL COMMENTS	Thank you for your careful revision. All previous comments have been addressed.